# Conversion of Marine Litter from Venice Lagoon into Marine Fuels via Thermochemical Route: The Overview of Products, Their Yield, Quality and Environmental Impact

**Gian Claudio Faussone** [1,2,*] **, Andrej Kržan** [2,3] **and Miha Grilc** [2,4,*]

1    Sintol, Corso Matteotti 32A, 10125 Torino, Italy
2    University of Nova Gorica, Vipavska 13, SI-5000 Nova Gorica, Slovenia; andrej.krzan@ki.si
3    Department for Polymer Chemistry and Technology, National Institute of Chemistry, Hajdrihova 19, SI-1000 Ljubljana, Slovenia
4    Department of Catalysis and Chemical Reaction Engineering, National Institute of Chemistry, Hajdrihova 19, SI-1000 Ljubljana, Slovenia
*    Correspondence: gcfaussone@lazabila.it (G.C.F.); miha.grilc@ki.si (M.G.); Tel.: +39-117790061 (G.C.F.); +38-61-4760540 (M.G.)

**Abstract:** Plastics floating in ocean gyres are a popular topic within pollution discussion; however, no simple solution exists to deal with marine litter. Overcoming limitations in collection, and perhaps even more in the environmentally, technically and economically acceptable use of the collected material, is of paramount importance. This paper presents initial results from converting plastic marine litter processed as-is, without pretreatment, and sorting into marine gas oil (MGO) compliant with the ISO8217 DMA standard via a pyrolysis and distillation process. Yields, composition and key properties of products along with levels of eight environmental contaminants potentially generated by the process are presented. More than 100 kg of actual marine litter from the Venice Lagoon, including polyolefins packaging and polyamides fishing nets, were converted into products at approximately 45 wt% yield of which approximately 50% (V/V) was MGO. By our knowledge, this is the first report of chemical recycling of real marine litter targeting the production of standardized marine fuels beyond laboratory scale, outlining coarse but realistic figures finally available as an initial benchmark. The process supports the concept of circularity in the blue economy and could be employed to tackle difficult terrestrial plastic waste to help prevent marine litter generation.

**Keywords:** marine litter; marine fuel; pyrolysis; circular economy; environmental impact

## 1. Introduction

### 1.1. Generalities on Marine Litter

According to UNEP (United Nations Environmental Program), marine litter is defined as "any persistent, manufactured or processed solid material discarded, disposed of or abandoned in the marine and coastal environment", but since plastic accounts for the majority of persistent litter items in the sea, special attention has to be paid to this type of waste; according to the Ellen MacArthur Foundation, there could be more plastic than fish in the oceans by 2050 [1]. Plastic debris has been found in every coast, sea surface and sea bottom [2], accumulating in every environment [3,4]. Since all plastic is generated on land, its presence in seas and oceans is attributed primarily to waste mismanagement: unappropriated disposal or recycling lead to waste going into rivers and eventually in the seas: the case of the Danube River is an example [5]. Jambeck et al., in the paper "Plastic waste inputs from land into the ocean" [6] estimate that from 4.8 to 12.7 Mt of plastic entered into the oceans in the year 2010 alone, and figures show a growing trend [7]. Eriksen et al. [8] estimates that more than 5 trillion plastic pieces weighing over 250.000 tons are already afloat at sea, but this represents just those polymers that float, while the marine litter lying on bottoms is not included in the figure and it is estimated that 70% of marine

litter sinks to the sea bottom with unknown consequences [9]. Regardless of exact figures, plastic waste in the seas and oceans undergoes a process of fragmentation leading to the formation of small debris that are ingested by living organisms, eventually entering into the food chain [10], harming ecosystems and human health; [11,12]. Once macroplastics are fragmented into micron-size debris, recovering and recycling seem almost impossible; the only solution besides preventing waste going into waters is to recover macroplastics before fragmentation occurs.

### 1.2. Recycling of Marine Litter

We can easily argue that marine litter is difficult to collect and with more effort, we can understand that it is even more difficult to recycle. In fact, standard recycling methods employed for specific terrestrial waste are based on mechanical reuse and recompounding, assuming homogeneity of the polymer family and relative cleanness [13]. However, those methods are technically ineffective and economically unfavorable because marine plastic debris is mixed, incrusted with organic matter and contaminated by salts. Since the first publications on marine litter in the late 1960s, the number of papers dealing with this topic has grown significantly, but still no specific research has been conducted on recycling [14]. Besides landfilling, incineration is the method most widely employed to treat marine debris [15], and only in very few countries, such as South Korea, marine litter is used to manufacture refuse derived fuel (RDF) [16], an energy carrier for selected end users. Other methods to deal with marine litter are basically economic instruments [17] where disincentives, incentives and penalties are used to induce a general behavior towards the prevention of marine litter generation. Since all litter that ends in the oceans is generated on land and considering that waste mismanagement is the primary cause of this pollution process, and ideal link exists between terrestrial mixed plastic mismanagement and marine litter. Therefore, effective technological solutions developed for marine litter recycling could be also employed to tackle the terrestrial mixed plastic waste stream before it becomes a problem, and thus preventing, or at least reducing to a great extent, waste mismanagement, which is the primary origin of marine litter generation.

### 1.3. Thermochemical Route to Recycle Marine Litter

Starting from this point, new pragmatic and interdisciplinary ways to recycle mixed waste plastic along with value generation is the key to addressing long-term marine litter [18]. The chemical route to recycle mixed feedstock might be a good option instead [19–21]. At its broadest definition, chemical recycling basically aims to recover the constituent building blocks of the feedstock as useful products. Within this pragmatic concept, the recovered building blocks can become valuable feedstock for the chemical industry or can become flexible energy carriers suitable for straight application displacing conventional oil-based fuels. In addition, manufactured products can be directly used by the same stakeholders involved in the depollution activities, for example, the fishermen who already catch plastic waste along with fish during their day-by-day activity. This is considered a strong incentive towards an inclusive approach to the problem solving: the operators of the maritime economy can thus become positive actors and trigger a virtuous depollution cycle, as also suggested elsewhere [22]. Pyrolysis is a method that can potentially achieve the required degree of marine litter decomposition and the use of additives and catalysts can drive this process towards desired outcomes. Several papers have been published on the pyrolysis of plastic with the goal to produce fossil fuel substitutes [23–25]; however, most are mainly academic studies difficult to compare to each other or to scale-up because of the equipment used [26,27]. Thus, very few data are reported on actual large-scale industrial applications. Previous work shows that it is possible to manufacture fuels meeting international ISO standards from landfill plastic by means of pyrolysis at 10 ton/day, a relatively large scale [28]; even ultralow sulfur fuels can be produced by mean of conventional petroleum industry technologies from waste plastics for terrestrial transportation purposes [29]. Interestingly, based on our knowledge, no data are

reported in literature about the pyrolysis of marine litter "as-is" for marine fuel synthesis; hence, our present research work is quite novel in this respect. Since huge amounts of fuel are used globally every year for marine transportation (207 Mt in 2017 alone, and over 36 Mt in 2019 in EU countries [30]), this makes a sustainable entry point for marine litter recycling products within a circular economy concept. In addition, often neglected is the environmental impact of the pyrolysis process itself, especially gaseous emissions into the atmosphere is the primary concern and contributes to the difficult public acceptance of such treatment plants. Thus, providing some results from a relative large-scale testing rig is of great significance to contribute to building confidence in the system and establishing the way for future investigation. Therefore, by performing trials on significant amounts of actual marine litter and by outlining coarse, but realistic figures of product yield, product quality and environmental impact, the fundamentals of marine litter recycling become finally available as an initial benchmark for further development. Within the activities of the EU funded project "Mapping and recycling of marine litter and ghost nets on the sea floor" [31], portable pyrolysis and distillation equipment were designed and operated. Since this project's focus was on marine litter and engagement of stakeholders involved in the maritime economy, the target products were marine fuels, which would be readily available for use in fishing boats.

## 2. Materials and Methods

### 2.1. Pyrolysis and Distillation Setup

The experimental set-up schematically shown in Figure 1 replicates, at smaller scale and with simplified components, an industrial plant producing fuels from plastic waste mined from a landfill, described elsewhere (Faussone, 2018). It is basically composed of the following functional units:

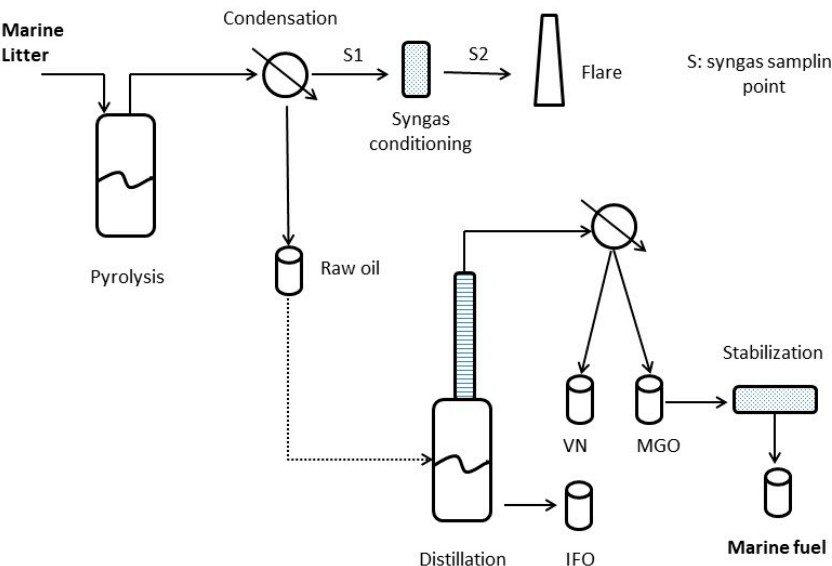

**Figure 1.** Experimental set-up composing the mobile prototype used.

1. A 100 L pyrolysis reactor, electrically heated and batch operated, directly connected with condensation train and raw pyrolysis oil collection tank. After the collection tank, the noncondensable syngas is conditioned before being flared adiabatically.
2. A 50 L atmospheric distillation unit to fractionate pyrolysis oil into the useful target products: a light distillate boiling up to 170 °C (virgin naphtha: VN), the middle distillate boiling up to 320 °C (marine gasoil: MGO) and the intermediate fuel oil (IFO), which is collected as residue from the distillation flask.

### 2.2. Monitoring of Environmentally Critical Substances

The noncondensable syngas was conditioned before being flared with the aim to remove potentially harmful substances from the syngas stream before it is flared, rather than from the flue gas. For this purpose, a state-of-the-art gas filtration system consisting of an alkaline water gurgle box followed by an in-line innovative polymer-engineered granular sorbent material (Adiox, patent Babcox and Wilcox, Gothenburg, Sweden) were employed. Two syngas sampling points (S1 and S2 in Figure 1) were used to monitor the presence of potential acidic and harmful precursor substances in the uncondensed syngas before and after the conditioning section. The environmental impact of the process was evaluated by monitoring the amount of selected pollutants in the syngas stream and by checking the efficiency of the conditioning. Based on previous work and by following Italian legislation, special attention was paid to substances that can lead to the formation of particularly harmful effluents. In our case, halogens, particularly chlorine, are of great importance because hydrochloric acid is formed from PVC thermal degradation [32]. It is known that hydrochloric acid in the syngas stream can lead to formation of dangerous dioxins if not properly flared [33]. Nielsen et al. [34] studied the formation of toxic gases such as hydrogen cyanide, carbon monoxide and ammonia during pyrolysis of polyacrylonitrile and nylons. Sulfur compounds were monitored because sulfur contamination from organic origin is likely to occur in untreated marine litter. Pentane was monitored to check the presence of combustible gas and thus determine if the flare could self-sustain the combustion in the absence of support combustible gas and potentially be exploited for power generation. Eight critical substances were thus monitored during process operation: pentane, hydrogen sulfide, hydrochloric acid, sulfur dioxide, ammonia, hydrogen fluoride, carbon monoxide and hydrogen cyanide.

Monitoring of pollutants was performed using color-reaction sampling vials (Drager, Lübeck, Germany). This method allows a precise, direct and easy ppm-range measurement in the field, and is widely employed by control authorities and industry for routine process monitoring. Since the results are immediate, process control is also possible owing to this method. Sampling was performed manually by a dedicated manual pump with a cycle counter. Vials were selected for maximum sensitivity and to avoid interference between different substances.

### 2.3. Marine Litter Collection

The marine litter used in the experiments was collected manually on the selected area of the Venice Lagoon and by scuba divers on the sea bottom. After 8 hours of natural drying, it was processed "as-is" without any pretreatment or special sorting operation. The litter was not chemically characterized; however, from visual inspection the composition was generally plastic containers and bottles and light packaging wrapping. Other constituents included fisheries and aquaculture litter, for example, plastic nets from mussel farming, nylon fishing gears and nets, litter from onboard fish and ichthyic products management such as expanded polystyrene boxes and their components, and other waste from aquaculture activities, for example, mooring lines, rubber gloves, boots, boat components such as cover sheets. Some of the marine litter used is portrayed in Figure 2. This is confirmed in literature where Galgani et al. [35] report that plastic is the most important part of marine litter, sometimes being 100% of the floating litter, and Consoli et al. [36] indicate that benthic marine litter is composed of fishing gear (32%) and general plastic (68%). Only in deep seafloor (30–300 m depth) fishing gear represent the dominant part of debris (89%) [37]. Since post-consumer plastic waste is largely composed of packaging made from just three polymer families: PE, PP and PET [38], mussel nets from PP, and the fishing nets are mainly made of nylons (6 and 6.6), we can assume that the marine litter being pyrolized was largely made of polyolefins, polyesters and polyamides with some contribution of polystyrene and rubbers. Therefore, we can expect positive contributions on oil yields from polyolefins, counterbalanced by lower yields from polyamides and polyesters [23,24].

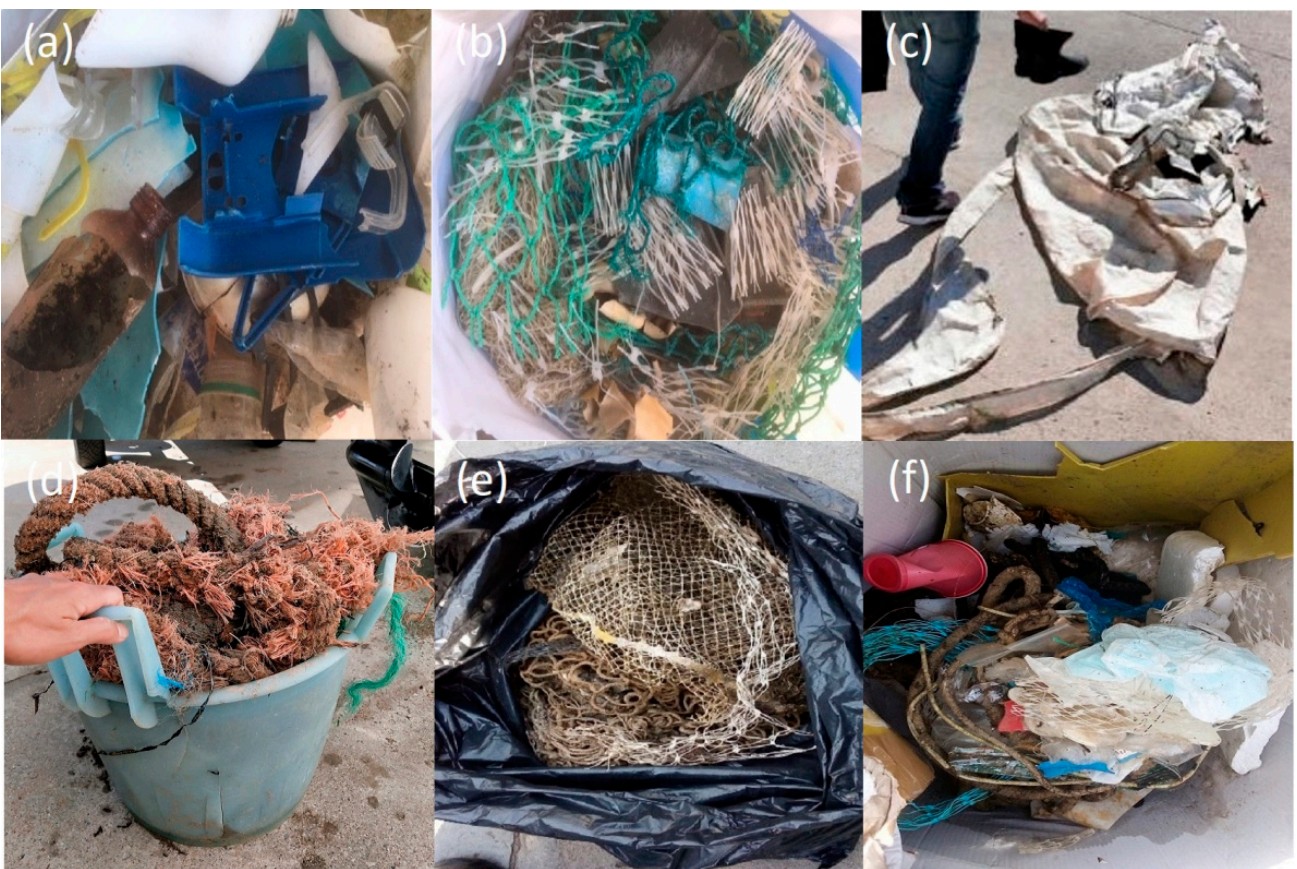

**Figure 2.** Actual marine litter used in pyrolysis tests: (**a**) mixed containers, (**b**) mussel nets, (**c**) ecoleather, (**d**) mooring lines, (**e**) trawl fishing nets, and (**f**) mixed floating litter.

Clearly not all collected material was plastic litter. Items such as boat cover debris, metal cans, dirt, organic matter (algae and seaweed), sea sand and mussel residues were present. Only macroscopic nonplastic items were removed when possible (metal cans and glass bottles). Scrap automobile tires were also present due to a widespread use as boat fenders. Scrap tires, albeit also collected during the project, were not included in the distillation experiments since it is easily recognizable and already investigated elsewhere [39].

*2.4. Marine Litter Processing and Product Characterization*

Representative samples of actual marine litter were processed by mean of pyrolysis. After loading, the pyrolysis reactor was closed and electric heaters were employed to maintain wall temperature at 450 ± 20 °C from the beginning of the cycle until its end after 4 h time. This cycle duration was chosen because in previous test runs no remaining condensed liquids were observed after 4 h. Evolving vapors were allowed to flow through the condensation train owing to the autogenic pressure. When the cycle was terminated the heaters were switched off and the reactor was left cooling overnight and then manually cleaned.

Inorganic additives based on CaO commonly employed in concrete manufacturing were added to the feedstock in fixed amount in order to control the pyrolysis process toward desired outcomes. The pyrolysis oil was then distilled and fractionated into three products. Finally, the distilled products were conditioned by means of sorbents routinely employed in the refinery industry to shape the final characteristics. The IFO was evaluated as per compliance with ISO8217:2017 Residual fuel classification, the MGO was evaluated as per compliance with ISO8217:2017, DMA, Distillate fuel classification. MGO was also analyzed by gas chromatography with a flame ionization detector (GC-FID) (2010 Ultra, Shimadzu, Kyoto, Japan). Compounds were successfully separated using a Zebron ZB-

5MS capillary column (60 m × 0.25 mm × 0.25 μm, Phenomenex, Torrance, CA, USA) and identified by a mass spectrometer and retention indexes. Undiluted sample (0.2 uL) was injected (using the AOC-20i autosampler) into a preheated (280 °C) split–splitless injector (split ratio of 120 and column flow rate of 1.5 mL/min were set) and further separated at 80 °C for 6 min followed by gradual heating to the final temperature of 310 °C, which was maintained for 10 min. A simulated distillation profile was obtained based on the GC-FID peak areas at their retention indexes and corresponding boiling points defined in NIST 17 library. GC results for MGO were compared with commercially available terrestrial diesel fuel. VN was not deeply investigated in this work since evaluation of feedstock for olefin conversion was outside the scope of this work. The amount of solid residue in the form of char was significant, and an even higher quantity of residue was found, considering the inorganics addition. Characterization and potential uses of this material, however, are not discussed in this work.

The IFO and MGO fuels produced were analyzed according to ISO 8217:2017 prescribed standard methods for marine fuels [40], except for oxidation stability, which was determined by the DIN EN 16091:2012 method. The VN was evaluated by the determination of the Research Octane Number (RON) and Measured Octane Number (MON) according to DIN EN ISO 5164:2014 and DIN EN ISO 5163:2014, respectively.

## 3. Results and Discussion

### 3.1. Product Yields

A number of pyrolysis runs using marine litter collected from the Venice Lagoon in 2020 were performed. ML loads between 4.0 and 11.5 kg were used, with the total waste processed reaching close to 100 kg. Table 1 shows the oil yield from each trial performed with marine litter only and one trial with scrap tire. The amount of recovered residue and calculated amount of gas is also reported for completeness; Table 2 summarizes the output from the distillation of the pyrolysis oil. All pyrolysis oil was homogenized prior to distillation in order to provide the overall mean composition. The difference between the mass of recovered oil and the initial mass of feedstock was recovered as solid residue and not condensable gas, the latter being flared after conditioning. The targeted products were MGO (Marine Gas Oil), IFO (Intermediate Fuel Oil) and VN (Virgin Naphtha).

**Table 1.** Summary of pyrolysis of marine litter tests.

| Trial # | ML kg | Oil kg | Residue (a) kg | Gas (b) kg | Yield Oil wt% | Yield Res. wt% | Yield Gas wt% |
|---|---|---|---|---|---|---|---|
| 1 (ML + nets) | 9.0 | 5.7 | 1.7 | 1.6 | 63.3 | 18.9 | 17.8 |
| 2 (ML + nets) | 6.6 | 3.2 | 1.8 | 1.6 | 48.5 | 27.3 | 24.2 |
| 3 (largely nets) | 6.3 | 1.0 (c) | 2.7 | 2.6 | 15.9 | 42.9 | 41.3 |
| 4 (sunk ML) | 5.0 | 2.1 | 1.7 | 1.2 | 42.0 | 34.0 | 24.0 |
| 5 (sunk ML) | 5.0 | 2.4 | 1.0 | 1.6 | 48.0 | 20.0 | 32.0 |
| 6 (sunk ML) | 5.0 | 2.6 | 1.0 | 1.4 | 52.0 | 20.0 | 28.0 |
| 7 (sunk ML) | 7.5 | 4.0 | 3.2 | 0.3 | 53.3 | 42.7 | 4.0 |
| 8 (ecoleather) | 9.0 | 1.8 | 5.5 | 1.7 | 19.6 | 61.1 | 19.3 |
| 9 (sunk ML) | 7.5 | 3.6 | 1.0 | 2.9 | 48.0 | 13.3 | 38.7 |
| 10 (sunk ML) | 11 | 5.2 | 2.3 | 3.5 | 47.3 | 20.9 | 31.8 |
| 11 (ML) | 11.5 | 7.2 | 2.0 | 2.3 | 62.6 | 17.4 | 20.0 |
| 12 (rubbers) | 6.0 | 1.9 | 3.5 | 0.6 | 32.0 | 58.3 | 9.7 |
| 13 (ML + rubber) | 6.2 | 1.6 | 1.2 | 3.4 | 25.8 | 19.4 | 54.8 |
| 14 (floating) | 4.0 | 2.4 | 0.5 | 1.1 | 60.0 | 12.5 | 27.5 |
| **TOT ML** | **99.6** | **44.7** | **29.1** | **25.8** | **44.9** | **29.2** | **25.9** |
| Scrap tire | 3.0 | 0.8 | 1.3 | 0.9 | 26.7 | 43.3 | 30.0 |
| **ML incl. tire** | **102.6** | **45.5** | **30.4** | **26.7** | **44.3** | **29.6** | **26.1** |

(a) Net of additives; (b) calculated by difference; (c) recovered water not included.

**Table 2.** Product yield collected from cumulative pyrolysis oil (feed).

|  | Feed | VN | MGO | IFO | $H_2O$ | Losses |
|---|---|---|---|---|---|---|
| Volume (L) | 34 | 8 | 17.5 | 2 | 3.6 | 2.9 |
| Fraction (vol%) | 100.0% | 23.5% | 51.5% | 5.9% | 10.6% | 8.5% |

Pyrolysis of different polymers types clearly result in different products and yields. For example, from Blazso et al. [24], it is known that pyrolysis of PE between 420 and 500 °C lead to straight chain hydrocarbons in the C1 to C20 and higher range. Pyrolysis of PP under the same conditions gives more volatile oil than PE. PVC undergoes a step of dehydrochlorination and benzene formation; generally, oil yield is low due to the greater molar presence of chlorine in relation to that of carbon and hydrogen. Pyrolysis of PET leads to formation of terephthalic acid with hydrolysis being the main decomposition route [41] and nylon-6 could be recovered as caprolactam via thermal decomposition [42]. Clearly, processing all of these polymers together results in oil yield and quality that are prone to great variability. Therefore, we should consider each of the single reported trials as part of a larger single pyrolysis experiment, where a larger quantity processed leads to a better estimation of an average, large-scale result. For this reason, we processed approximately 100 kg of marine litter with each single experiment carried out under the same conditions.

As shown in Table 1, oil yield varied significantly with a maximum of almost 63% in run #1 and a minimum of almost 16% in run #3 where an additional 1 L of water was collected and separated. This great difference may be explained by the marine litter used during the trials. For example, because there was no pretreatment and no presorting, for trial #8 a large piece of "ecoleather" (see Figure 2) coated with unknown resin was used. "Ecoleather" is not even made of plastic, instead being either cotton fabric or natural leather with small amount of coating for waterproofing. Obviously, the actual amount of polymer available to undergo thermal degradation was significantly lower compared to the weighed litter. Nevertheless, the plastic portion was successfully converted into oil. On the other hand, high yields of trials #1 and #11 were attained by using unsorted plastic marine litter, whose composition was still unknown but clearly recognizably made of polyolefin-based plastic. Both trials #1 and #3 also used fishing nets as feedstock and some water was found in the product but easily visible only for trial #3. Since fishing nets are predominantly made of nylon-6 and nylon-6.6, water formation is possible. In fact, despite the type of nylon present in the load was unknown, we can argue that water formation comes mainly from nylon-6.6, which was also responsible for low yield in trial #3. Czernik et al. [43] studied the catalytic pyrolysis of nylon-6 to recover caprolactam with a bench-scale fluidized bed reactor system, reporting an oil yield of 85% and no water formation. Similar results were reported by Kim et al. [44]; researchers investigated the pyrolysis of fresh and waste fishing nets made of nylon-6 without catalyst at 440 °C and reported that the carbon number distribution of the pyrolysis oil was random, but again without water formation. Finally, in their review of thermal decomposition of aliphatic nylons, Levchik et al. [42] reported that diacid-diamine type nylons (such as nylon 6.6) produce mostly linear or cyclic oligomeric fragments with a significant number of secondary products such as water and CO coming from adipic acid fragmentation.

Coming to the products of distillation, MGO is the dominant outcome, but also high amount of VN has to be considered as a valuable product. Interestingly, only about 6% (V/V) of IFO was obtained, therefore this approach leads to the production of fuels (and potentially also chemicals for the VN) with the highest economic value among the marine fuel family members. Since distillation was performed at atmospheric pressure, cracking reactions occurred leading to product loss in form of gas and coke. This suggests that vacuum distillation could be more appropriate for pyrolysis oil production to reduce cracking reactions and related product loss. Presence of water, originally not recognizable in the pyrolysis oil, probably because in form of emulsion was this time recovered together

with the VN and easily separated. Presence of water was not problematic for distillation, but since water is not miscible with hydrocarbons, a sort of "steam current distillation" was actually occurring, at least in the beginning phase. Since boiling temperatures change in presence of water, this aspect is also of some importance if low carbon number products are desired to be separated selectively and not as bulk as done in this case. Figure 3 shows the visual appearance of the products.

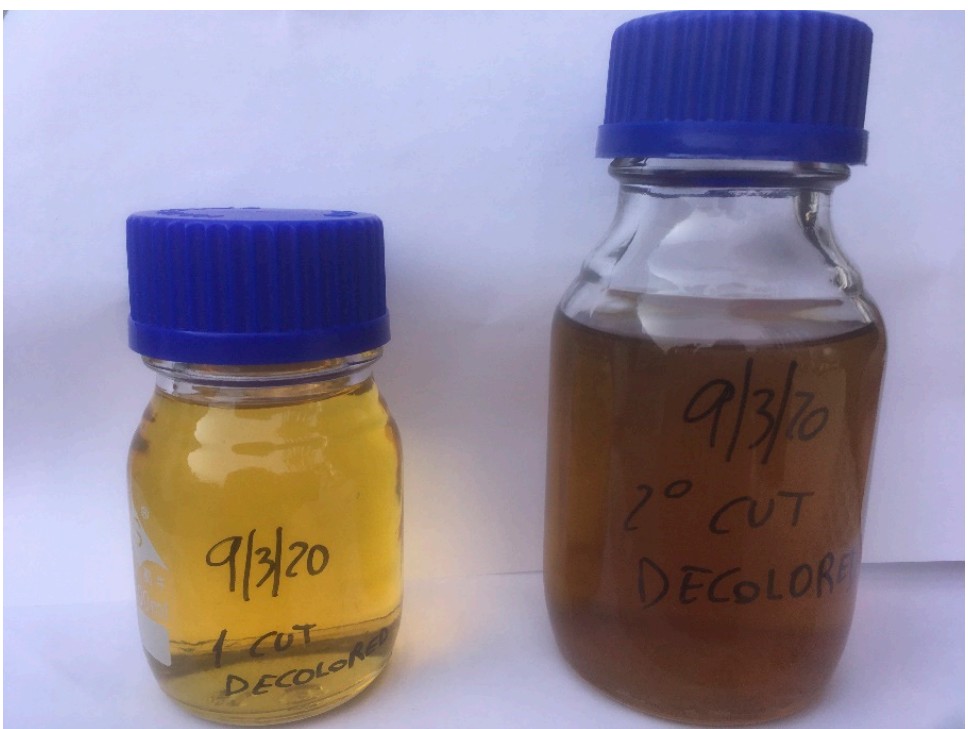

**Figure 3.** Visual appearance of VN (**left**) and MGO (**right**).

### 3.2. Product Quality

Table 3 reports the analytical values for the IFO. Kinematic viscosity, density, cetane index, sulfur content, flash point, hydrogen sulfide and acid value allow this fuel to be classified as DMB (also termed "marine diesel oil") according to ISO8217:2017. However, still the pour point of 34 °C is above the limit of 6 °C, ash content of 0.667 % (m/m) is above the limit of 0.010 % (m/m) and water content is slightly above the limit of 0.30% (V/V) to meet DMB specs. Among these parameters, clearly the cold flow properties are those that characterize this product the most. Pour point is, however, only slightly above the limit for ISO-F residual marine fuels category (limit of 30 °C for RMD, RME, RMG, RMK) and so is the water content of 0.58 vs. limit of 0.5 % (V/V). In this case, the content of sodium (maximum limit of 100 mg/kg) and aluminum plus silicon (maximum limit 60 mg/kg for RMK) are the greatest obstacles to overcome for a full classification of this product as residual marine fuel. Sodium likely comes from sea salt contamination, but it becomes problematic in fuels when a significant amount of vanadium is also present. Sodium acts as a paste for vanadium slag, creating a eutectic mixture that melts within the operating temperature of combustion in engine cylinders [45]. Since presence of vanadium is very low in the present case, below 0.5 mg/kg, this problem of slagging is unlikely to occur, but further testing is required to confirm this hypothesis.

**Table 3.** Analytical results of IFO.

| Parameter | Value | Unit |
|---|---|---|
| Kinematic viscosity (50 °C) | 8.61 | $mm^2/s$ |
| Density (15 °C) | 841.2 | $kg/m^3$ |
| Cetane index | 70.6 | - |
| Sulfur content | 34 | ppm |
| Flash point | >100 | °C |
| Hydrogen sulfide | <2 | ppm |
| Acid value | 0.04 | mg KOH/g |
| Sediment content | 1.58 | wt% (m/m) |
| Carbon residue | 1.86 | wt% (m/m) |
| Pour point | 34 | °C |
| Water content | 0.58 | vol% |
| Ash content (775 °C) | 0.667 | wt% |
| Vanadium (V) | <0.5 | mg/kg |
| Sodium (Na) | 959 | mg/kg |
| Aluminum (Al) | <0.5 | mg/kg |
| Silicon (Si) | 130 | mg/kg |
| Calcium (Ca) | 13.8 | mg/kg |
| Zinc (Zn) | 1.4 | mg/kg |
| Phosphorus (P) | 48.2 | mg/kg |
| Cloud point | >20 | °C |
| HFRR (Lubricity at 60 °C) | 210 | μm |

In contrast, the low content of sulfur, only 34 ppm, is much better than the low sulfur marine fuel requirement of 1000 ppm set by IMO MARPOL annex VI [46] for SOx air pollution prevention within Emission Control Areas (ECAs), and by far better than the 5000 ppm limit set for outside ECAs. This limit applies in the four established ECAs: the Baltic Sea, the North Sea and the North American area for January 2020; the number of ECAs is predicted to grow in the future. Therefore, this product could be of some interest for blending with traditional residual fuels to help meet environmental targets, or as fuel for heating applications. Another possibility would be to upgrade produced IFO into wax or petrolatum, a not a fuel product, for chemical applications.

Table 4 reports the analytical values for MGO. In this case, all parameters except the flash point are within the limits set for DMA classification. This low value of the flash point is clearly due to incomplete separation of the low temperature boiling fraction; since analysis shows that 10% of the product is still recovered at 178 °C, the distillation phase was improved. Therefore, a second distillation with cut temperature of 180 °C was performed and the flash point was raised to 58 °C, only 2 °C below the required 60 °C. For further application, instead of raising the cut temperature, a better separation column would attain the same result. Albeit the content of sodium is not regulated for DMA, analysis shows this amount is now below 0.5 mg/kg and since vanadium content is also below 0.5 mg/kg, slag formation during combustion is not an issue. Oxidation stability was determined by a different method from the one prescribed by ISO8217:2017 and therefore the result is not comparable. However, after three months of storage no change in color or evident sign of sludge precipitation was observed.

Other values are by far better than the prescribed ones, cetane index of 61 is significantly higher than the minimum value of 40. This is a great accomplishment: since higher cetane fuels have shorter ignition delay, fast engines (>500 rpm) can also use this type of fuel. This also implies that higher efficiency and hence lower fuel consumption is expected from the use of this fuel compared to standards, benefitting both the environment and economy. When sulfur content is considered, the value of 196 mg/kg is by far better than the lowest limit of 1000 mg/kg set by IMO for the ECAS and qualifies this product as ULSFO "ultralow sulfur fuel oil". Since standard MGO sold on the market has a maximum sulfur content of 1500 ppm and IMO2020 grade bunker has the maximum limit of 5000 ppm, MGO produced is also better than commercial and could extend *de facto*

ECAs standards coverage with evident benefits for the environment. Interestingly, the measured acid value of 0.136 mg KOH/g is far inferior to the limit of 0.5 mg KOH/g. This fact is noticeable because presence of polyethylene terephthalate (PET), clearly present in the marine litter as liquid bottles, is often problematic during pyrolysis because PET decomposes into sublimating terephthalic and benzoic acids, which cause issues in the processing facilities and acidify the condensate [47,48]. Cold flow properties are in line with specification and the pour point of −6 °C would qualify this fuel as DMA "winter quality".

**Table 4.** Analytical results of MGO after homogenization of all products listed in Table 1.

| Parameter | Result | Unit |
|---|---|---|
| Kinematic viscosity (50 °C) | 1.848 | $mm^2/s$ |
| Density (15 °C) | 802.9 | $kg/m^3$ |
| Cetane index | 61.3 | - |
| 10% (V/V) recovery | 178.3 | °C |
| 50% (V/V) recovery | 257.1 | °C |
| 90% (V/V) recovery | 347.0 | °C |
| Sulfur content | 196 | ppm |
| Flash point | 35.0 | °C |
| Flash point (improved) | 58.0 | °C |
| Hydrogen sulfide | <2 | ppm |
| Acid value | 0.136 | mg KOH/g |
| Sediment content | 0.02 | % (m/m) |
| Carbon residue | <0.10 | % (m/m) |
| Pour point | −6 | °C |
| Water content | 0.01 | % (V/V) |
| Ash content (775 °C) | <0.001 | % (m/m) |
| Vanadium (V) | <0.5 | mg/kg |
| Sodium (Na) | <0.5 | mg/kg |
| Aluminum (Al) | 0.5 | mg/kg |
| Silicon (Si) | 113 | mg/kg |
| Calcium (Ca) | <0.5 | mg/kg |
| Zinc (Zn) | <0.5 | mg/kg |
| Phosphorus (P) | 4.2 | mg/kg |
| Cloud point | 14 | °C |
| HFRR (Lubricity at 60 °C) | 240 | μm |
| Oxidation stability | 15.46 | min |

As already mentioned, the low temperature boiling distillate (VN) was not fully characterized in this work because gasoline is not a common fuel for fishing boats and because this feedstock was selected for a different application. However, for completeness, at least the RON and MON were analyzed, giving values of 68.8 and 65.3 respectively. Values are far from the minimum values of 95 and 85 for RON and MON, respectively, as prescribed by EN228:2012 [49], for example, and therefore if employed in spark-ignited engines, this product would lead to "knocking" issues. Since the VN corresponds to the gasoline distilling range, low values can be explained by the presence of relative high amount of low octane hydrocarbons such as hexane and heptane counterbalanced by aromatics.

Gas chromatography analysis shown in Figure 4a reveals that MGO mainly consists of linear, unbranched hydrocarbons with the chain length distribution within the range of C7 and C29, with the largest presence of n-pentadecane. All of the hydrocarbons appear as pairs of saturated n-alkanes and their corresponding n-alk-1-enes (linear, unbranched alkenes with a double on their first carbon atom). The alk-1-ene to alkane ratios (of 1:2) in the pairs gradually decrease and become negligible for longer chains. Hydrocarbons are unbranched as they are predominantly formed from linear polyethylene in the marine litter during the pyrolysis, while the radical stabilization predominantly forms a double bond on the shorter pyrolysis/cracking fraction. The comparative GC-FID chromatogram for (terrestrial) diesel fuel shows a similarly wide but less symmetrical distribution of

hydrocarbons (dodecane is the most abundant compound). However, diesel does not contain any alkenes but comprises considerable amounts of branched isomers in between the n-alkane peaks, which also results in a much smoother simulated distillation profile (Figure 4b) in comparison to MGO, where discrete alkane/alkene pairs result in stepwise profile. MGO-simulated distillation profile generally matches well the experimentally determined data shown in Table 4; specifically, the simulation predicts 10% recovery at 190 °C, 50% recovery at 282 °C and 90% recovery at 369 °C.

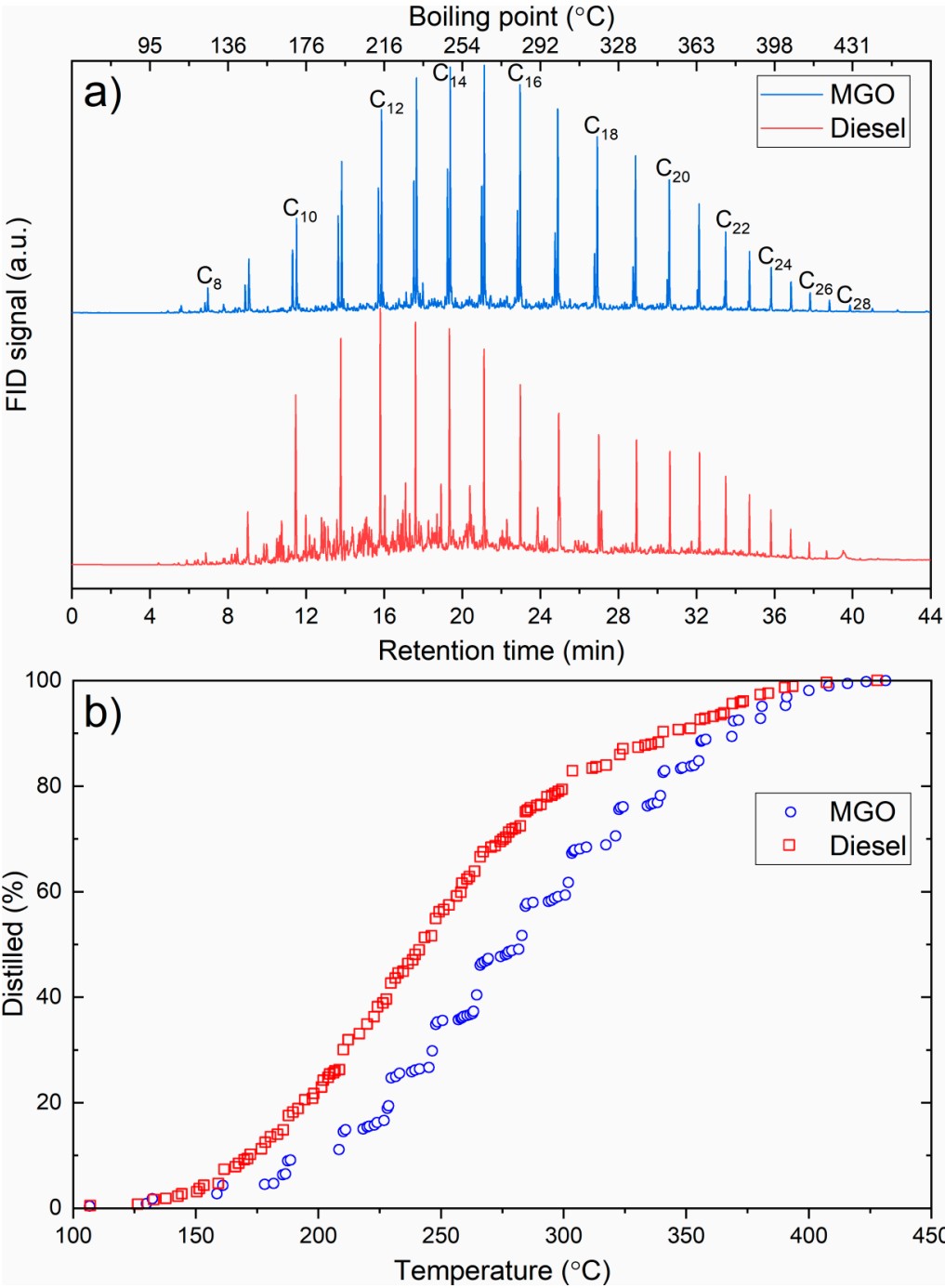

**Figure 4.** GC-FID results for MGO and commercially available diesel fuel: (**a**) chromatograms with estimated boiling point of eluted compounds ($R^2$ = 0.997 correlation between retention time and boiling point), (**b**) GC-FID-based simulated distillation profiles.

### 3.3. Pollutants in Noncondensed Gas

Pollutants levels are reported in Table 5. Pentane and carbon monoxide were always at high levels before and after the conditioning step. This is a positive indication since both are combustible gases and can self-sustain the combustion even without support fuel or can be exploited for power generation or other energy uses. Pentane is likely to come from thermal decomposition of polyolefins while carbon monoxide can either come from decarboxylation reactions or thermal decomposition of organic contaminants such as seaweeds. Hydrochloric acid was not detected; therefore, either it was not present in the feedstock, such as PVC, for example, or successfully removed during pyrolysis. Hydrogen cyanide was present in the syngas before conditioning, but was successfully removed in the process. Ammonia level dropped greatly from 600 to approximately 100 ppm after conditioning, and possibly even lower levels could be achieved by further optimization. Ammonia and hydrogen cyanide are likely to come from polyamide decomposition and have to be considered as normal pollutants to manage when applying pyrolysis for marine litter conversion to fuels. No traces of hydrogen sulfide and hydrogen fluoride were detected and sulfur dioxide was also detected at a very low level. Since the heterogeneity of marine litter is high, specific pollutant precursor substances are also unlikely to be found in great concentrations; therefore, we can consider these results a good base for future dedicated research on emission control systems. In addition, a state-of-the-art filtration system similar to those of municipal solid waste incinerators was used, proving that existing technologies can be also employed for pyrolysis of marine litter.

**Table 5.** Pollutant concentrations in the syngas before (S1) and after conditioning (S2).

|  | S1 | S2 |
| :---: | :---: | :---: |
| **Pollutant** | **ppm** | **ppm** |
| Pentane $C_5H_{12}$ | >1500 | >1500 |
| Hydrogen sulfide $H_2S$ | <0.2 * | <0.2 * |
| Hydrochloric acid HCl | <0.2 * | <0.2 * |
| Sulfur dioxide $SO_2$ | n.a. | <<0.5 |
| Ammonia $NH_3$ | >600 | 100 |
| Hydrogen fluoride HF | <0.5 * | <0.5 * |
| Carbon monoxide CO | >700 | >700 |
| Hydrogen cyanide HCN | 5 | 0 |

* Reading was 0; the value indicated value is the limit of detection.

## 4. Conclusions

Marine litter is a growing problem worldwide with limited remediation instruments. Despite the familiar images of plastic floating in oceans and seas, most of it is sunk and a significant amount of fishing nets are present. Due to the heterogeneity of the litter and the presence of significant contaminants (e.g., biofouling and sand), classical recycling methods based on mechanical processing are not viable options and a chemical route is preferred. In addition to preventing new litter flowing into the oceans, a new pragmatic, not idealistic approach to manage heterogeneous plastic waste is needed to trigger a prevention and depollution cycle. In this work, about 100 kg of actual marine litter, including fishing nets, was processed "as-is" by means of pyrolysis and successfully largely converted, approximately 51% (V/V), to marine gasoil (MGO) compliant with ISO8217 DMA standard by mean of distillation. Due to the extreme heterogeneity and organic contamination of marine litter, its composition was largely unknown but still somehow identified as being predominantly polyolefin and polyamides. Clearly, this led the oil yield to extend over a wide range, with an average of 45% (m/m) being 16% (m/m) and 63% (m/m) of the threshold values. These results were reached even without sorting, indicating that yield could be improved by a simple feedstock triage, for example, eliminating those elements that are clearly not of plastic origin and should follow another disposal route. This result suggests that a drop-in, low sulfur-containing fuel for maritime transportation can be

produced via this approach. Hence, fishermen and other stakeholders of the blue economy can find a direct and tangible incentive to align the environmental objective with their daily activity. Data collected on eight potentially harmful contaminants show that environmental impact can be efficiently managed and curbed to safe levels if proper process conditions and additives are used. These coarse but realistic numbers provide base data for marine litter conversion to marine fuels via pyrolysis, which are now available as an initial benchmark for further development, with the hope that, owing to a pragmatic, not idealistic approach, the prediction of finding more plastic than fish in the oceans by 2050 will be wrong.

**Author Contributions:** Conceptualization, G.C.F.; methodology, G.C.F. and M.G.; investigation, G.C.F.; writing—original draft preparation, G.C.F. and A.K.; writing—review and editing, G.C.F. and M.G.; visualization, G.C.F. and M.G.; supervision, M.G.; project administration, G.C.F. and M.G.; funding acquisition, G.C.F. and M.G. All authors have read and agreed to the published version of the manuscript.

**Funding:** This research has received funding within the marGnet project from the European Union's EASME–EMFF funding program—Sustainable Blue Economy Call, under agreement n. EASME/EMFF/2017/1.2.1.12/S2/05/SI2.789314. Slovenian Research Agency funding is also acknowledged within the research programme P2-0152 and research project J2-2492.

**Institutional Review Board Statement:** Not applicable.

**Informed Consent Statement:** Not applicable.

**Data Availability Statement:** Data is contained within the article.

**Conflicts of Interest:** The authors declare no conflict of interest.

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
