# Peer review of "Conversion of Marine Litter from Venice Lagoon into Marine Fuels via Thermochemical Route: The Overview of Products, Their Yield, Quality and Environmental Impact"

_sustainability, doi:10.3390/su13169481_

Round 1

Reviewer 1 Report

The manuscript entitled" Conversion of Marine Litter Rom Venice Lagoon into Marine Fuels Via Thermochemical Route: The Overview of Products, Their Yield, Quality and Environmental Impact" particularly deal with the thermochemical conversion of marine litters via pyrolysis to marine fuel. Though this technique is quite old, however it might be possible not applied on Marine litter yet as waste utilization. The manuscript needs major revision in terms to highlight the novelty of the research. My comments are as follows: 

  1. Abstract needs to improve with major finding on process and waste material characterization. 
  2. Introduction is too long, should be reduced and only relevant information and literature should be included. Authors are advised to included the literature on various types of Marine litters and their segregation and possible effects on product yield in terms on pyrolysis. 
  3. Authors might need address the issue with organic and non-organic marine waste. 
  4. Results and discussion needs major revision in order to compare the present study with literature on pyrolysis.
  5. Authors are advised to include the GCMS or GCFID results for confirmation of the products/fuels produced. 

Reviewer 2 Report

This is a very interesting study describing the pyrolysis of real marine litter for the production of marine fuels. My comments are below:

  1. The authors made a very informative Introduction, which is appreciated. However, I suggest revising it, e.g., into more paragraphs, to increase the readability.
  2. More information for the pyrolysis experiment is needed, such as the reactor heating rate and the detailed experiment procedure.
  3. What are the state-of-the-art gas filtration system and innovative sorbent materials employed for the flue gas treatment?
  4. Inorganic additives commonly employed in concrete manufacturing were added to the feedstock. What are these materials? Are they acting as the catalysts?
  5. The authors presented a very detailed property of the derived fuels, which is the aim of this work. It would be more interesting to attract more readers like the referee if the GC-FID/MS results could be included to show the chemical compositions of the liquid products.

Round 2

Reviewer 1 Report

The manuscript is revised carefully, I recommend publication in present form.